# Anomalous Elastic Properties of Attraction-Dominated DNA Self-Assembled 2D Films and the Resultant Dynamic Biodetection Signals of Microbeam Sensors

**DOI:** 10.3390/nano9040543

**Published:** 2019-04-03

**Authors:** Junzheng Wu, Ying Zhang, Nenghui Zhang

**Affiliations:** 1Shanghai Key Laboratory of Mechanics in Energy Engineering, Shanghai Institute of Applied Mathematics and Mechanics, Shanghai University, Shanghai 200072, China; wujunzheng123@126.com (J.W.); zhangxiaoyi@shu.edu.cn (Y.Z.); 2Department of Mechanics, College of Sciences, Shanghai University, Shanghai 200444, China

**Keywords:** DNA film, micromechanical biosensor, elastic property, natural frequency, multiscale method

## Abstract

The condensation of DNA helices has been regularly found in cell nucleus, bacterial nucleoids, and viral capsids, and during its relevant biodetections the attractive interactions between DNA helices could not be neglected. In this letter, we theoretically characterize the elastic properties of double-stranded DNA (dsDNA) self-assembled 2D films and their multiscale correlations with the dynamic detection signals of DNA-microbeams. The comparison of attraction- and repulsion-dominated DNA films shows that the competition between attractive and repulsive micro-interactions endows dsDNA films in multivalent salt solutions with anomalous elastic properties such as tensile surface stresses and negative moduli; the occurrence of the tensile surface stress for the attraction-dominated DNA self-assembled film reveals the possible physical mechanism of the condensation found in organism. Furthermore, dynamic analyses of a hinged–hinged DNA-microbeam reveal non-monotonous frequency shifts due to attraction- or repulsion-dominated dsDNA adsorptions and dynamic instability occurrence during the detections of repulsion-dominated DNA films. This dynamic instability implies the existence of a sensitive interval of material parameters in which DNA adsorptions will induce a drastic natural frequency shift or a jump of vibration mode even with a tiny variation of the detection conditions. These new insights might provide us some potential guidance to achieve an ultra-highly sensitive biodetection method in the future.

## 1. Introduction

Unlike the wormlike genomic DNA in dilute solutions, DNA condensation has been regularly found in cell nucleus, bacterial nucleoids, and viral capsids [1,2,3]. In the condensed state, despite the strong electrostatic repulsion that exists between negatively charged molecules, DNA double helices are locally aligned and separated by just one or two layers of water [1,4], and this indicates the emergence of the attractive interactions induced by multivalent cations, lipids, or polymers [1,3]. Several theoretical works such as attractive electrostatic forces, screened Debye–Hückel interactions, and water-structuring or hydration forces, have tried to explain the physical origin of the attractive interactions [4]. However, the lack of experimental measurements prevented further development and discrimination among these alternative theories [4]. Recently, by the single-molecule experiments using biochemical, osmotic stress, X-ray scattering, optical techniques, and silicon nanotweezers integrating with a microfluidic device, the three-dimensional condensation of DNA in solution has been studied [5,6]. Also, Langevin dynamics simulations have been used to study the DNA condensation in single-molecule experiments [2]. Furthermore, experiments have shown that the structures of Mg cation with deep-ion-binding sites and phosphoester sites make it capable of bridging features, not only along the helix, but also across helix binding [7].

Surface-effect-based nanomechanical biosensor is a unique tool for measuring biomolecular interactions and molecular conformational changes without molecular labeling [8,9,10]. For instance, as a promotion of the general observation of three-dimensional aggregation of DNA in solution, Mertens et al. provided an alternative method to obtain the direct information about the forces involved in a two-dimensional condensation of DNA by using functionalized DNA- microcantilever sensors [5]. Experiment results give direct evidence that trivalent ions turn the repulsive electrostatic forces between short strands of single-stranded DNA into attractive as a previous step to condensation [5]. Other works also show that different kinds of buffer salt solutions [5], salt concentrations [11], DNA packing densities [11], and environment temperatures [12] will trigger the change of surface stress and the resultant transition of bending direction. Eom et al. revealed that the resonant frequency shift for a microcantilever resonator due to biomolecular adsorption depends on, not only the mass of adsorbed biomolecules, but also the biomolecular interactions [13]. Lee et al. observed an anomalous increase in the resonant frequency during the Au adsorption on the microcantilever, and speculated that the positive frequency shift was ascribed to the variation in the spring constant related to the surface stress [14]. Tamayo et al. also showed that the adsorption position and the thickness ratio between the adsorbed layer and the microbeam induced an anomalous resonant frequency shift [15]. However, the quantified assessment description of the relationship between the anomalous signals and the experiment conditions, especially for the attraction cases, still remains an open question.

Different from the previous analysis of piezoelectric properties of double-stranded DNA (dsDNA) films and its effect on the static detection signals of microcantilevers [16], this paper is devoted to the establishment of a multiscale model to characterize the macroscale elastic properties of dsDNA films and their correlations with the anomalous dynamic detection signals of hinged–hinged microbeams induced by micro-interactions. First, two mesoscopic potentials of free energy for a repulsion-dominated dsDNA film in NaCl solution or attraction-dominated dsDNA films in multivalent salt solutions are used to predict their elastic properties, including surface stress and elastic modulus. The comparative study of attraction- and repulsion-dominated DNA films shows that the competition between attractive and repulsive micro-interactions endows the attraction-dominated dsDNA films with anomalous elastic properties such as tensile surface stress and negative modulus, and the predicted tensile surface stress reveals the possible physical mechanism of the condensation found in organism. Next, the first-order natural frequency shifts of a hinged–hinged microbeam with a repulsion- or attraction-dominated DNA film are discussed. Numerical results show a non-monotonic variation in frequency shifts due to dsDNA adsorptions and totally different responses between detections of attraction-dominated films and that of repulsion-dominated films, and the dynamic instability occurs during the detections of repulsion-dominated DNA films. This instability indicates that there is a sensitive interval of material parameters in which DNA adsorptions will induce a drastic natural frequency shift or a jump of vibration mode from stability to instability even with a tiny variation of the detection conditions. At last, the physical mechanism underlying these non-monotonous variations in detection signals of dsDNA films at different experiment conditions is discussed.

## 2. Multiscale Analytical Model

In this paper, through the energy method, we are looking forward to establishing a multiscale analytical model to describe the relationship between the surface elastic properties of adsorbed DNA films and the detection signals of DNA-microbeam systems. 

Figure 1a shows the scheme of the Atomic Force Microscope (AFM) measurement for biodetections [17], in which a laser is used to capture the adsorption induced deflection of the microcantilever and its reflection is collected by a quadrant photodetector or by a position sensitive detector (PSD). The structure and the relevant coordinate of the microbeam are shown in Figure 1b. We will investigate a symmetric adsorption with advantages of minimizing both the effects of thermal drift and non-specific binding interactions with the backside of the microcantilever [18,19]. The structure consists of three layers: The two symmetric adsorbed DNA films and the SiN*_x_*/Si substrate with the length of *l* and the width of b. And *E*_p_ and *E*_s_, and *h*_p_ and *h*_s_ represent the elastic moduli and thicknesses of the self- or directed-assembled DNA film and substrate, respectively. The *x*-axis is established at the geometric midplane of the substrate, and the positive direction of the *z*-axis points to the bottom film.

### 2.1. Elastic Properties of Adsorbed DNA Films 

In this section, the adsorbed DNA film is treated as an elastomer. According to continuum mechanics, if the free energy of the self- or directed-assembled DNA film is derived, its elastic properties in a uniaxial compressive/tensile state can be easily obtained as [20]
(1)$$E_{p}=3\eta \partial^{2}W_{b}/\partial \varepsilon^{2}{|}_{\varepsilon =0},\;\sigma_{p}=3\eta \partial W_{b}/\partial \varepsilon {|}_{\varepsilon =0}$$
where *E*_p_ is the elastic modulus, *σ*_p_ is the surface stress, *ε* is the axial strain, *η* is the DNA packing density, and $\eta =2/(\sqrt{3}d_{0}^{2})$ for the hexagonal packing pattern, in which *d*_0_ is the initial interaxial distance [21,22]; *W*_b_ is the free energy per unit length between two parallel DNA cylinders. However, there is no a unified formula for the free energy of DNA solutions. In the following section, two mesoscopic free energy potentials will be, respectively, introduced for a repulsion-dominated dsDNA film in NaCl solution or attraction-dominated dsDNA films in multivalent salt solutions.

As for the mesoscopic free energy of dsDNA in multivalent salt solutions, such as spermine [H_2_N(CH_2_)_3_NH(CH_2_)_4_NH(CH_2_)_3_NH_2_] (valence +4), Co(NH_3_)_6_Cl_3_ (valence +3) and sp^6+^ [H_2_N(CH_2_)_3_NH(CH_2_)_3_NH(CH_2_)_3_NH(CH_2_)_3_NH(CH_2_)_3_ NH_2_] (valence +6), by combining the single-molecule magnetic tweezers and osmotic stress experiments, Todd et al. separated the attractive and repulsive components from the total intermolecular interactions, and proposed an alternative interaction potential of free energy [4]. The free energy per length is given as
(2)$$W_{b1}=\Delta G_{\mathrm{rep}}+\Delta G_{\mathrm{att}}=\sqrt{3}\lambda (d+\lambda /2)C_{R}e^{-2d/\lambda }/2-\sqrt{3}\lambda (d+\lambda )C_{A}e^{-d/\lambda },$$
where $\Delta G_{\mathrm{rep}}$ and $\Delta G_{\mathrm{att}}$ represent the repulsive and attractive interaction potentials, respectively. By convention, the repulsive interaction potential is defined as positive, and the attractive potential is negative. *λ* = 4.6 Å is the decay length, *C*_R_ and *C*_A_ are the corresponding prefactors related to the specific salt conditions, and *d* is the interaxial distance. According to our previous models [21], the interaxial distance, *d*, between parallel DNA cylinders after microbeam bending, is given as $d=(1+\varepsilon )d_{0}$, in which *d*_0_ is the initial interaxial distance, and *ε* is the bending strain. The thickness of the adsorbed film is approximately taken as the contour length of DNA chain [21,23], namely, $h_{p}\approx Na$, *N* is the DNA nucleotide number, *a* is the nucleotide length directly obtained from STM experiment, and *a* = 0.34 nm for dsDNA [24].

As for the mesoscopic free energy of dsDNA in NaCl solution, based on a liquid-crystal model and osmotic pressure experiments, Strey et al. [25] presented a repulsion-dominated interaction potential which has been used to effectively predict the deflection and surface stress of DNA-microbeam systems. The repulsive interaction energy per unit length between two parallel DNA cylinders is given as
(3)$$W_{b2}=W_{e}+W_{h}+W_{c},$$
where *W*_e_, *W*_h_, and *W*_c_, are, respectively, electrostatic energy, hydration energy, and configurational entropy, and
$$W_{e}(z,d)=a_{0}\exp(-d/\lambda_{D})/\sqrt{d/\lambda_{D}},\;W_{h}=b_{0}\exp(-d/\lambda_{H})/\sqrt{d/\lambda_{H}},$$
$$W_{c}=c_{0}k_{B}Tk_{c}^{-1/4}\sqrt[{4}]{\partial^{2}(W_{e}+W_{h})/\partial d^{2}-(1/d)\partial (W_{e}+W_{h})/\partial d},$$
where *λ*_D_ is the Debye screening length, *λ*_H_ is the correlation length of water [25], *a*_0_, *b*_0_, and *c*_0_ are the fitting parameters for DNA interactions; $k_{B}$ is the Boltzmann constant, *T* is the temperature, $k_{c}=k_{B}Tl_{p}^{\mathrm{ds}}$ is the bending stiffness of a single-molecule dsDNA chain, $l_{p}^{\mathrm{ds}}$ is the persistence length of dsDNA, $l_{p}^{\mathrm{ds}}$= (50 + 0.0324/*I*) nm, and *I* is the buffer salt concentration [26].

Finally, substituting *W*_b1_ in Equation (2) or *W*_b2_ in Equation (3) into Equation (1) yields the elastic modulus and surface stress of the adsorbed dsDNA film in multivalent or monovalent NaCl solutions.

### 2.2. Natural Frequency of DNA-Microbeam System 

This section is dedicated to investigating the influence of DNA elastic properties on the natural frequency of microbeam. The governing equation of the DNA-microbeam system will be established by using the energy method, and the first-order variation of the relevant generalized Hamiltonian function is written as
(4)$$\delta \int_{t_{1}}^{t_{2}}\int_{0}^{l}(T-\Pi )dxdt+\int_{t_{1}}^{t_{2}}\int_{0}^{l}\delta Vdxdt=0$$
where *T*, *∏*, and *V*, respectively, represent the kinetic energy per unit axial length, total elastic potential energy of the DNA-microbeam system, and external work per unit axial length; *t*_1_ and *t*_2_ are different moments.

As for the dynamic response of a hinged–hinged beam, the kinetic energy per unit axial length, *T*, can be written as
(5)$$T=(m+\Delta m){\left(\partial w/\partial t\right)}^{2}/2,\;$$
where *m* and Δ*m* represent the linear mass density of the substrate and the DNA film, respectively. 

Considering the surface stress $\sigma_{p}$ as a symmetric external load along the surface of the substrate, the external work per unit axial length can be written as
(6)$$V=\sigma_{p}h_{p}b{\left(\partial w/\partial x\right)}^{2}.$$

The total elastic potential energy of the DNA-microbeam system, *∏*, includes three parts: The elastic potential energy stored in the substrate, $W_{s}$, the effective elastic potential energy of the top DNA film, $W_{p,\mathrm{top}}$, and that of the bottom DNA film, $W_{p,\mathrm{bot}}$, i.e.,
(7)$$\Pi =W_{s}+W_{p,\mathrm{bot}}+W_{p,\mathrm{top}},\;$$
in which
$$\begin{array}{l} W_{s}=bl\int_{-h_{s}/2}^{h_{s}/2}E_{s}\varepsilon^{2}dz/2\\ W_{p,\mathrm{bot}}=bl\int_{h_{s}/2}^{h_{s}/2+h_{p}}(\sigma_{p}\varepsilon +E_{s}\varepsilon^{2}/2)dz\\ W_{p,\mathrm{top}}=bl\int_{-h_{s}/2-h_{p}}^{-h_{s}/2}(\sigma_{p}\varepsilon +E_{s}\varepsilon^{2}/2)dz \end{array}$$
where the bending strain ε can be described by Zhang’s two variable method [27], i.e., $\varepsilon =\varepsilon_{0}-\kappa z$, where *κ* is the curvature of the neutral axis and $\varepsilon_{0}$ is the normal strain along the *x*-direction at *z* = 0. The effective elastic potential energies of the adsorbed DNA films are estimated by using Equations (1)–(3).

Substituting Equations (5)–(7) into Equation (4), the vibrational differential equation is obtained as
(8)$$(E_{s}I_{s}+\Delta EI)\partial^{4}w/\partial x^{4}+2\sigma_{p}h_{p}b\partial^{2}w/\partial x^{2}=(m+\Delta m)\partial^{2}w/\partial t^{2}$$
where *m* = *ρ**bh*_s_, *ρ*, and *E*_s_*I*_s_ are, respectively, the effective linear mass density, the mass density, and the stiffness of the substrate; *b* is the beam width; Δ*m* ≈ 2*ηbN* × 1.6 × 10^−21^/1600 kg is the effective mass of the DNA film per unit axial length of the beam [28]; Δ*E**I =*
$E_{p}bI_{u,\mathrm{bot}2}+E_{p}bI_{u,\mathrm{top}2}$ is the additional stiffness induced by DNA adsorptions, and $I_{u,\mathrm{bot}2}=I_{u,\mathrm{top}2}=\int_{h_{s}/2}^{h_{s}/2+h_{p}}z^{2}dz$. Note that the effective stiffness could reduce to that of Eom et al. [13], Wang and Feng [29], and Lu et al. [30] in the case of tiny film thickness.

The separation variable method is used to solve Equation (8). Assume *w*(*x*, *t*) = *Φ*(*x*)*q*(*t*), where *Φ*(*x*) is the modal function and *q*(*t*) is the time domain function. To illustrate the surface effects, here only the hinged–hinged microbeam is considered. Substituting the above solution form into Equation (8) yields the following *i-*th mode natural frequency of the beam after DNA adsorptions, i.e.,
(9)$$\begin{array}{l} p_{i}=p_{i}^{0}\sqrt{\alpha_{1}\alpha_{2}\alpha_{3}}\\ \alpha_{1}=1+\Delta EI/E_{s}I_{s},\\ \alpha_{2}=1+2\sigma_{p}h_{p}bl^{2}/[i^{2}\pi^{2}(E_{s}I_{s}+\Delta EI)],\\ \alpha_{3}=1-\Delta m_{\mathrm{DNA}}/(\rho ah_{s}+\Delta m_{\mathrm{DNA}}), \end{array}$$
where $p_{i}^{0}$ is the *i-*th mode natural frequency without surface effect; *α*_1_, *α*_2_, and *α*_3_ are the dimensionless parameters standing for the effects of surface stiffness, stress–stiffness coupling, and additional mass, respectively. Obviously, the above three effects are closely related to the geometric and elastic properties of adsorbed DNA films and the substrate.

To summarize, different microscopic interactions of surface molecules may endow DNA films with totally different mechanical properties, which are closely relevant to the complex detection signals of DNA-microbeams. With the above analytical model, we can quantify these multiscale correlations between macroscopic detection signals and surface elastic properties of the adsorbed film induced by microscopic molecular interactions.

## 3. Results and Discussion

In computation, dsDNA nucleotide number is taken as *N* = 25, the substrate size *l* = 9 μm, and *b* = 0.4 μm for dynamic analyses of a hinged–hinged microbeam. Due to the length-to-width ratio of the substrate, the biaxial modulus is taken as *E*_s_/(1−*μ*_s_), where elastic modulus *E*_s_ = 180 GPa, and Poisson’s ratio *μ*_s_ = 0.27. The parameters in Equation (3) for dsDNA in 0.1 M NaCl solution is taken as: *a*_0_ = 0.41 × 10^−9^ J/m, *b*_0_ = 1.1 × 10^−7^ J/m, *c*_0_ = 0.8, *λ*_D_ = 0.974 nm, and *λ*_H_ = 0.288 nm [25]. Substituting the experimental data on Δ*G*_rep_ and Δ*G*_att_ = *W*_b2_
*−* Δ*G*_rep_ of Todd et al. [4] into Equation (2), we could obtain the prefactors *C*_R_ and *C*_A_ of Δ*G*_rep_ and Δ*G*_att_ for dsDNA, and the related parameters in different salt solutions are shown in Table 1. Here, 12.3 pN = 1*k*_B_*T*/*a*, in which the nucleotide length *a* = 0.34 nm. According to the previous osmotic pressure experiments [31], the interaxial spacing of dsDNA inside virus is about 2.6 nm, so the packing density can approximately reach 1.7 × 10^17^ chains/m^2^ for the hexagonal packing pattern.

First, we will study the variation of surface elastic properties of adsorbed dsDNA films and its mechanism induced by micro-interactions. By using Equation (1), the variation tendencies of surface stress with the packing density in several salt solutions are compared in Figure 2a. By convention, the positive value represents the compressive stress while the negative value represents the tensile one. In NaCl solution, the surface stress always behaved compressive and its value increased with the enhancement of the packing density. In addition, Figure 2b shows that the collected contributions of electrostatic energy, hydration energy, and configurational entropy led to the variation of surface stress, which was also the deformation mechanism of micro-beam sensor in NaCl solution.

Whereas in multivalent solutions (sp^6+^ and spermine), the surface stress exhibited different trends with nonmonotonic variations, as shown in Figure 2a. Taking sp^6+^ as an example, when the packing density *η* < 1.45 × 10^17^ chain/m^2^, the surface stress behaved tensile and this revealed the possible physical mechanism of the condensation found in organism induced by the attractive interactions between DNA helices; when *η* ≈ 1.2 × 10^17^ chain/m^2^, the tensile surface stress reached its maximum value, which provided us an opportunity to prepare a more sensitive sensor by the directed-assembled technique; when *η* ≈ 1.45 × 10^17^ chain/m^2^, the surface stress turned to be zero and the sensor might have lost any signals, and this is the most miserable situation in biodetections; when *η* > 1.45 × 10^17^ chain/m^2^, the surface stress behaved compressive inversely and this indicates the dominance of the repulsive interactions between DNA helices. Physically speaking, the competition between repulsive and attractive part of free energy make the surface stress changing from tensile to compressive, and this also interprets the mechanism of microbeam sensor deformation in sp^6+^ solution. As shown in Figure 2c, at a relatively low packing density, the dominance of contribution of the attractive part of the free energy resulted in tensile surface stresses; with the increase in the packing density, the repulsive part of the free energy gradually became more critical and eventually resulted in compressive surface stresses. However, the discrepancy in Co(NH_3_)_6_Cl_3_ came into sight. While in Co(NH_3_)_6_Cl_3_ solution, the surface stress will always be tensile. In addition, given the parameters exactly the same as in the experiment, the magnitude of the tensile surface stress was about 1 MPa, and it has the same order with Todd’s experimental result of osmotic pressure among DNA molecules, i.e., Π ∈ (0.1 MPa,10 MPa) [4].

Also, by using Equation (1), the variation tendencies of elastic modulus with the packing density have been studied. Figure 3a shows the elastic moduli of dsDNA films in various salt solutions. As we can see, with the similar tendency of the surface stress, the elastic modulus of the DNA film in NaCl solution always behaved positive and it increased with the enhancement of the packing density. As shown in Figure 3b, in NaCl solution, the collected contributions of electrostatic energy, hydration energy, and configurational entropy to the surface stress lead to the variation of elastic modulus at different packing densities. Nevertheless, the elastic modulus in a multivalent solution (sp^6+^, spermine, and Co(NH_3_)_6_Cl_3_) was negative at a relatively low DNA packing density, whereas it turned positive at a relatively high density. In addition, the elastic moduli in multivalent solutions were about one order of magnitude lower than that in NaCl solution. However, the critical packing densities in Figure 2a and 3a are different. For example, in sp^6+^ solutions, the DNA elastic modulus was negative when the packing density *η* < 1.14 × 10^17^ chain/m^2^, and became almost zero when *η* reached 1.14 × 10^17^ chain/m^2^, then turned positive when *η* > 1.14 × 10^17^ chain/m^2^. Also, there was a critical packing density for the negative modulus at *η* ≈ 0.95 × 10^17^ chain/m^2^. Furthermore, Figure 3c shows that the competition between the repulsive part and attractive part of free energy lead to the non-monotonic variation of elastic modulus. In addition, the magnitude of the elastic modulus of the DNA film in 0.1 M NaCl solution was about 0.1~100 MPa, which is similar to Zhang’s theoretical prediction [22], and slightly smaller than Legay’s (50 mM NaCl solution) [32], due to different salt concentrations and packing conditions as well as the inherent deficiency of AFM-based nano-indentation detection. What is more, our simulation showed a consistent monotonic trend with that of Domínguez’s theoretical predictions and approaches the order of their AFM experiment results [17]. It should be mentioned that negative elastic modulus is unstable in nature, however can be stabilized by lateral constraint [33,34]. As for the DNA film in the microbeam-based biosensor, it was actually pre-stretched during the immobilization process, namely restrained by the substrate. Figuratively speaking, imagining the DNA film as a pre-stretched spring, it is surely unstable without lateral constraint. When we dismiss the constraint and apply a tiny lateral tensile stress far less than the residual stress induced by pre-stretching, which is insufficient to remain the stable state, the pre-stretched spring will obviously be compressed and consequently behaves a negative modulus.

Next, by using Equation (9), we will study the variation of the natural frequency shift of a hinged–hinged microbeam induced by dsDNA adsorptions and its mechanism related to surface properties. As we can see from Equation (9), the natural frequency shift was the result of the competition between effects of surface stiffness, stress–stiffness coupling, and additional mass (*α*_1_, *α*_2_, *α*_3_), which is closely related to the elastic properties of adsorbed films induced by micro-interactions as well as the elastic and geometric properties of the substrate. It can be learned from the above discussions that, given the packing density *η* = 1.2 × 10^17^ chain/m^2^, the surface stress of dsDNA film will always behave compressive in NaCl solution or tensile in sp^6+^ solution, respectively. Considering the boundary constraints, obviously the substrate will be compressed in NaCl solution and stretched in sp^6+^ solution, respectively. Once the elastic moduli and surface stress of the adsorbed dsDNA film are known, the dynamic detection signals of dsDNA-microbeam could be easily obtained.

Figure 4 shows the first-order natural frequency shift of the hinged–hinged microbeam with the variation in the absolute value of film-to-substrate thickness ratio (i.e., *r* = |*h*_p_/*h*_s_|) and modulus ratio (i.e., *g* = |*E*_p_/*E*_s_|). First, as shown in Figure 4, the first-order natural frequency shift due to dsDNA adsorptions was mostly negative in NaCl solution and positive in sp^6+^ solution. Similar behavior has been discovered in Karabalin’s surface stress loaded beam experiments (beam length: 6 to 10 μm; width: 0.6 to 1 μm; thickness: 0.015 to 0.028 μm; Poisson’s ratio: 0 to 0.49) [35]) and Lachut’s analytical predictions [36]. Second, the amplitudes of the natural frequency shift in both solutions showed the similar tendency, namely, enhancing with the increase of the absolute value of film-to-substrate thickness ratio or modulus ratio. Actually, as shown in Figure 4, when the parameter values were relatively large, the stress–stiffness coupling effect *α*_2_ dominated the value of natural frequency shift. Taking sp^6+^ solution as example, given *r* = *g* = 0.04, the contributions of *α*_1,_
*α*_2,_
*α*_3_ to the first-order natural frequency shift were, respectively, −0.52%, 44.5%, and −0.6%, so the positive effect of the stress–stiffness coupling determined the upward trend of natural frequency shift.

Third, an anomalous invalid region is observed in Figure 4. Note that the DNA film in NaCl solution is in a repulsion-dominated state, and the microbeam vibrates in different modes depending on the specific experiment conditions:
(i)When the relation between the modulus ratio and the thickness ratio satisfies the following relation, $g\leq 9.016\times 10^{-6}/(6.22r^{3}-1.08\times 10^{-4}r^{2}-5.41\times 10^{-5}r)$, the microbeam vibrates in a linear phase, in which the frequency shift of a periodic vibration could be taken as an indication of DNA adsorptions; (ii)When their relation satisfies the following relation, $g>9.016\times 10^{-6}/(6.22r^{3}$$-1.08\times 10^{-4}r^{2}-5.41\times 10^{-5}r)$, the microbeam vibrates in a non-periodic way, which means a dynamic instability region (i.e., the anomalous blank area in Figure 4 appears.

It can be seen from the linear analytical solution to Equation (8) that, when the parameters locate at the condition (ii), the additional mass-relevant coefficient *α*_3_ is always greater than zero, the competition between the surface stress effect and the stiffness effect makes the signs of *α*_1_ and *α*_2_ opposite, and this means $p_{i}^{2}<0$, so its corresponding temporal-domain equation, $\ddot{q}(t)+p_{i}^{2}q(t)=0$, has a nonperiodic solution with $q_{1}(t)=-c_{1}p_{i}^{-2}e^{-p_{i}^{2}t}+c_{2}$; here, *c*_1_ and *c*_2_ are determined by the initial conditions. In other words, the motion increases exponentially. This is totally different from the linear periodic motion with $q_{2}(t)=c\sin(p_{i}t+\theta )$ when $p_{i}^{2}>0$, where *c* is also determined by the initial conditions. The restriction on the linear periodic motion endows the linear vibration natural frequency shift only with an upper limit of 100% in NaCl solution. The instability indicates the occurrence of a sensitive interval in which DNA adsorptions induce a drastic natural frequency shift even with a tiny variation of the detection conditions. Whereas the appearance of dynamic instability at condition (ii) will cause a sudden jump of vibration mode from stability to instability at the critical condition, and this means a relatively large deformation for the beam. In these cases, this dynamic instability might provide us a potential method to develop a ultra-highly sensitive detection method through the linear vibration-based material parameter controlling or a new nonlinear vibration-based technology in the future [37].

However, unlike the seemingly monotonicity observed in the global view as shown in Figure 4, when the parameter value was taken relatively small, the non-monotonic behavior came into sight in the local zoom view, as shown in Figure 5. Taking the detection of attraction-dominated films in sp^6+^ solution as an example, given the modulus ratio *g* = 0.04 in Figure 5a, when the thickness ratio *r* < 0.00572, the frequency shift was negative, and became almost zero when *r* reached 0.00572, then turned positive when *r* > 0.00572. Also, there was a critical value for the negative frequency shift at *r* ≈ 0.00347. However, given the modulus ratio *g* = 0.1, the frequency shift tendency in the detection of repulsion-dominated films as shown in Figure 5b was totally different from that of attraction-dominated films, and with the increase in thickness ratio the shift turned from positive to negative. In addition, the non-monotonic behavior observed in Figure 5b was negligible when the modulus ratio was relatively small (e.g., *g* = 0.04). The variation between positive and negative frequency shift has been found in DNA hybridization experiments by Zheng et al. [38], and similar anomalous non-monotonic tendencies have been found in the study of alkanethiol adsorption by Tamayo et al. (beam material: Si; critical thickness ratio *h*_p_/*h*_s_ approximates to 0.15) [15] and Au adsorption by Lee et al. (beam material: lead zirconate titanate (PZT); critical thickness ratio *h*_p_/*h*_s_ approximates to 0.000445) [14] when the adsorption layer is relatively thin compared with the substrate.

The physical mechanism underlying these non-monotonic responses of the microbeam to different detection conditions can be interpreted by the present multiscale analytical model. As we can see from Figure 5a, during the detection of the attraction-dominated films in sp^6+^ solution, the effects of additional mass *α*_3_ and surface stiffness *α*_1_ always behaved negatively whereas the effect of stress–stiffness coupling *α*_2_ behaved positively, which means that the stress–stiffness coupling effect dominated when the thickness ratio *r* > 0.00572, whereas both additional mass and surface stiffness effect played an essential role when *r* < 0.00572. In a word, the frequency shift of the microbeam is the result of the competition of the above-mentioned three effects closely related to the elastic and geometric properties of the adsorbed film and the substrate. Different surface elastic properties of repulsion-dominated films in NaCl solution leads to a totally different tendency in the frequency shift. These conclusions further verified the previous conclusion based on experimental observations that the stress–stiffness coupling effect becomes more dominant with the increase in the absolute value of film-to-substrate thickness ratio [14]. In addition, these non-monotonic variations and totally different responses in frequency shifts during the detections of attraction- or repulsion-dominated dsDNA films provide us an alternative perspective to promote the sensitivity of surface-effect-based biosensors.

It should be mentioned that, in the case of detecting the adsorbed DNA film with an anomalous negative elastic modulus, if we mistake it as a general material with a positive modulus, this might induce a large prediction error. Taking the dynamic signals of DNA films with *h*_p_/*h*_s_ = 0.003 in sp^6+^ solution as example, the elastic modulus of DNA film is about −1 MPa. As shown in Figure 5a, the original prediction of the frequency shift considering the negative elastic modulus was about −0.059%. However, if we take the elastic modulus as 1 MPa inversely, the frequency shift will be mistakenly estimated as 0.01%, and the relative prediction error between these two predictions is about 100%. In a word, this unneglectable prediction error indicates that the anomalous negative elastic modulus of the adsorbed DNA film has great influence on microbeam-based biodetection signals.

## 4. Conclusions

In this paper, we characterized the elastic properties of dsDNA films and established a multiscale analytical model to describe the relationship between the surface mechanical properties of DNA self-assembled 2D films and the detection signals of DNA-microbeam systems. The related predictions agree well with the AFM indentation experiment [17] and microbeam vibration experiment [14,35]. Analytical results show that the microscale attractive interactions between DNA chains will lead to anomalous negative elastic moduli and tensile surface stresses, and the occurrence of this tensile surface stress for the attraction-dominated DNA film reveals the possible physical mechanism of the condensation found in organism. In addition, the dynamic analysis of a hinged–hinged microbeam in multivalent salt solutions suggests that, despite the non-monotonic tendency of frequency shift when the absolute value of film-to-substrate thickness ratio is relatively small, above a critical film-to-substrate thickness ratio, an attraction-dominated film could always induce a positive natural frequency shift, totally different from the detection signal for a repulsion-dominated DNA film. These insights emphasize the importance of the stress–stiffness coupling effect in dynamic responses and provide us an alternative perspective to promote the sensitivity of surface-effect-based biosensor. What is more important, during the detection of a repulsion-dominated DNA film, dynamic instability appears after the critical conditions, which brings about a jump of vibration mode from stable to instable states with a relatively large displacement of a microbeam, and this indicates the existence of a sensitive interval in which DNA adsorptions will induce a drastic natural frequency shift even with a tiny variation of the detection conditions. In these cases, this dynamic instability might provide us a potential method to develop an ultra-highly sensitive detection method through the linear vibration-based material parameter controlling or a new nonlinear vibration-based technology in the future.

## Figures and Tables

**Figure 1 nanomaterials-09-00543-f001:**
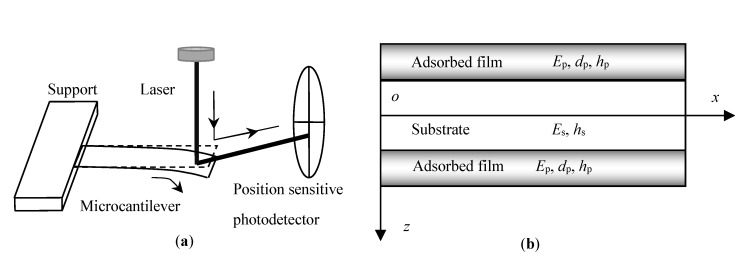
(**a**) Scheme of the Atomic Force Microscope (AFM) measurement; (**b**) schematic showing a microbeam and its coordinate system.

**Figure 2 nanomaterials-09-00543-f002:**
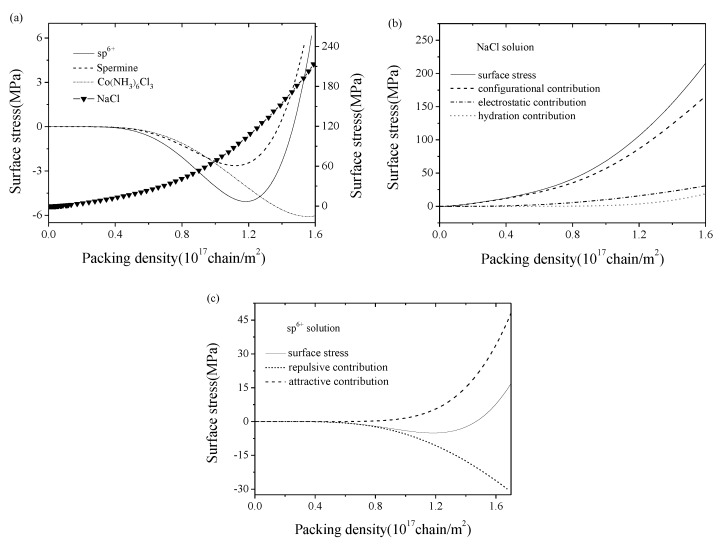
(**a**) Surface stress variation of dsDNA films with packing density in sp^6+^, spermine, Co(NH_3_)_6_Cl_3_, and NaCl solutions. The left longitudinal axis is related to DNA in multivalent solutions and the right one is related to DNA in NaCl solution. By convention, the positive value represents the compressive stress while the negative value means the tensile one. (**b**) Contributions of electrostatic energy, hydration energy, and configurational entropy to the surface stress in NaCl solution. (**c**) Contributions of the repulsive and attractive parts of free energy to the surface stress in sp^6+^ solution.

**Figure 3 nanomaterials-09-00543-f003:**
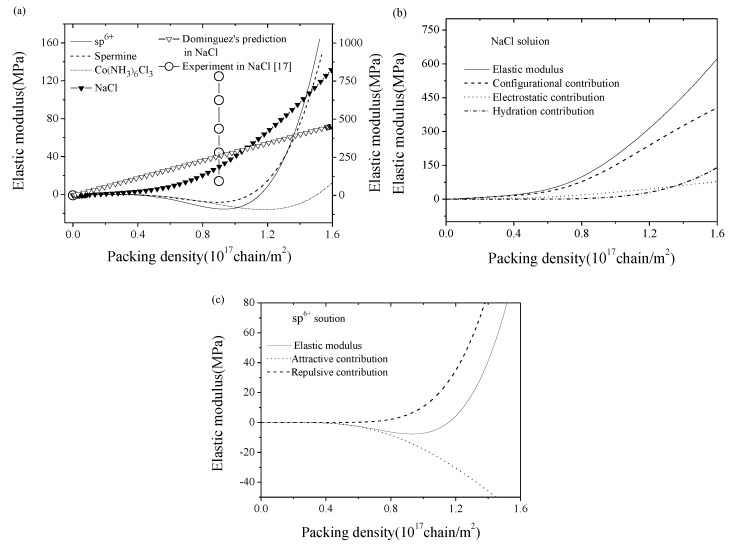
(**a**) Theoretical elastic modulus variation (lines) of dsDNA films with packing density in sp^6+^, spermine, Co(NH_3_)_6_Cl_3_, NaCl solutions, and Domínguez’s AFM experiment results [17] (circles) in NaCl solution. The left longitudinal axis is related to DNA in multivalent solutions and the right one is related to DNA in NaCl solution. (**b**) Contributions of electrostatic energy, hydration energy, and configurational entropy to the elastic modulus in NaCl solution. (**c**) Contributions of the repulsive and attractive part of free energy to the elastic modulus in sp^6+^ solution.

**Figure 4 nanomaterials-09-00543-f004:**
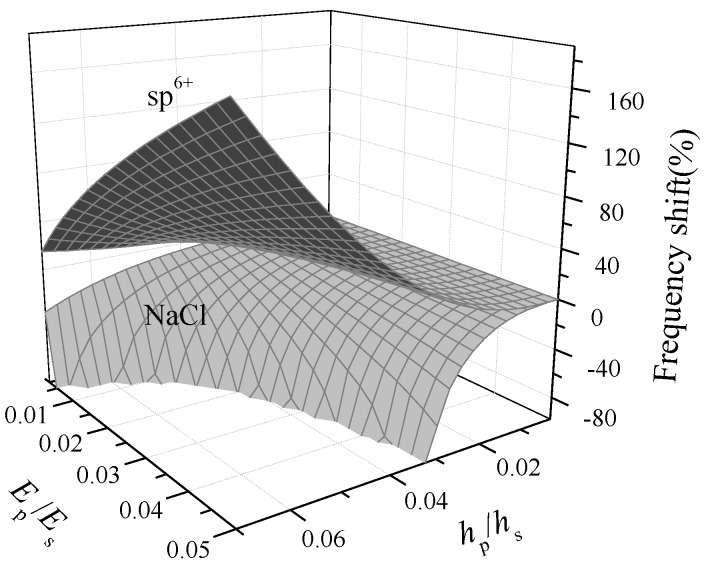
The first-order natural frequency shift of a hinged–hinged dsDNA-microbeam with the variation in the absolute value of film-to-substrate thickness ratio (i.e., *r =* |*h*_p_/*h*_s_|) and modulus ratio (i.e., *g* = |*E*_p_/*E*_s_|) in sp^6+^ and NaCl solutions when the packing density *η* = 1.2 × 10^17^ chain/m^2^.

**Figure 5 nanomaterials-09-00543-f005:**
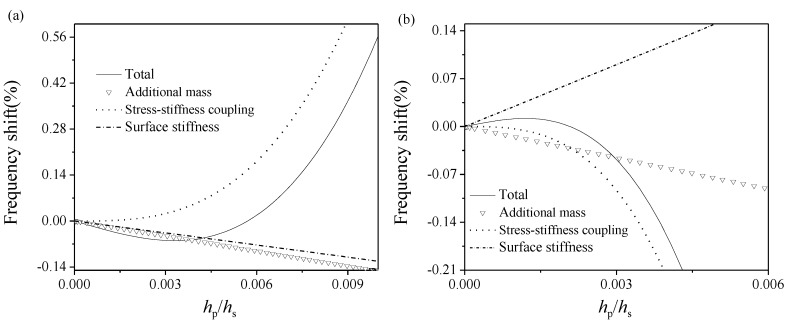
The first-order natural frequency shift of a hinged–hinged dsDNA-microbeam and contributions of surface stiffness (*α*_1_), stress–stiffness coupling (*α*_2_), and additional mass (*α*_3_) effects with the variation in the absolute value of film-to-substrate thickness ratio (i.e., *r =* |*h*_p_/*h*_s_|) in sp^6+^ and NaCl solutions when the packing density *η* = 1.2 × 10^17^ chain/m^2^. (**a**) sp^6+^ solution, film-to-substrate modulus ratio *g* = |*E*_p_/*E*_s_| = 0.04; (**b**) NaCl solution, *g* = |*E*_p_/*E*_s_| = 0.1.

**Table 1 nanomaterials-09-00543-t001:** Experimental data of double-stranded DNA (dsDNA) at different salt solutions [4] and the solved prefactors.

	*d*, Å	*aW*_b2_, *k*_B_*T/a*	*a*Δ*G*_rep_, *k*_B_*T/a*	*C*_A_, MPa	*C*_R_, MPa
Co(NH_3_)_6_Cl_3_	27.75	−0.21	0.17	755.83	303, 444
Spermine	28.15	−0.33	0.29	945.89	508, 714
sp^6+^	27.65	−0.38	0.39	1503.26	668, 743

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
