# Peer review of "Anomalous Elastic Properties of Attraction-Dominated DNA Self-Assembled 2D Films and the Resultant Dynamic Biodetection Signals of Microbeam Sensors"

_nanomaterials, 2019, doi:10.3390/nano9040543_

Reviewer 1 Report

The authors are kindly requested to format the manuscript as specified in the „Instructions for authors”.

Please check the English spelling, i.e. in the figures.

For the convenience of the readers, it is recommended to reorganize the manuscript.

Please check/give the name of all compounds mentioned in the manuscript.

All variables in the equations should be defined.

Provide a more detailed discussion about the anomalous elastic properties and the dynamic biodetection signals.

Author Response

Reviewer 1

The authors are kindly requested to format the manuscript as specified in the “Instructions for authors”.

Comment 1: Please check the English spelling, i.e. in the figures.

Response: We have modified our English spelling. (Line 143, 236, 253, 292, 365)

Comment 2: For the convenience of the readers, it is recommended to reorganize the manuscript.

Response: Thanks for the suggestion, we have reorganized our manuscript. (Section 2)

Comment 3: Please check/give the name of all compounds mentioned in the manuscript.

All variables in the equations should be defined.

Response: According to the comment, several variables have been defined. (Line 98, 144, 146, 209)

Comment 4: Provide a more detailed discussion about the anomalous elastic properties and the dynamic biodetection signals. 

Response: Thanks for the suggestion, we have made a more detailed discussion. (Line 388- 396)

Reviewer 2 Report

The paper by Zhang studies the electrostatic effects of cations on DNA mechanical properties. As such it is of current interest, as referenced there is growing body of work emerging from the Fujita group at Univ Tokyo in this area, which may explain the short comments of reviewer 2. Here the authors may wish to consider their results in terms of DNA length mechanics and inter-strand mechanics. This would very much be the case for the long sp6 systems which would be capable of bridging features along the helix but also of cross helix binding.

There is a solid body of X-ray structural data which could help, especially Mg cation structures which show both deep ion binding sites and also phosphoester sites which cross link helices. This point should taken up.

After dealing with the above structure based information the paper will be acceptable.

Author Response

Reviewer 2

CommentThe paper by Zhang studies the electrostatic effects of cations on DNA mechanical properties. As such it is of current interest, as referenced there is growing body of work emerging from the Fujita group at Univ Tokyo in this area, which may explain the short comments of reviewer 2. Here the authors may wish to consider their results in terms of DNA length mechanics and inter-strand mechanics. This would very much be the case for the long sp6 systems which would be capable of bridging features along the helix but also of cross helix binding.

There is a solid body of X-ray structural data which could help, especially Mg cation structures which show both deep ion binding sites and also phosphoester sites which cross link helices. This point should taken up.

After dealing with the above structure based information the paper will be acceptable.

Response: Thanks for the kind suggestion. According to the comments, we have made a few changes. (Line 44- 46)

However, although we have searched for more than 50 articles in the limited time, the significant articles of Fujita’s group in the area of DNA binding have yet been found. We would be much appreciated if the reviewer could provide us the specific information of the Fujita’s article and we will further quote to the relevant studies.

Round  2

Reviewer 1 Report

The authors addressed properly to all of the reviewer's comments.

Reviewer 2 Report

Correction carried out OK to publish

Paper to help with their request

Elucidating the mechanism of the considerable mechanical stiffening of DNA induced by the couple Zn(2+/)Calix[4]arene-1,3-O-diphosphorous acid

By: Tauran, Yannick; Tarhan, Mehmet C.; Mollet, Laurent; et al.

SCIENTIFIC REPORTS  Volume: 8     Article Number: 1226   Published: JAN 19 2018

Accept

This manuscript is a resubmission of an earlier submission. The following is a list of the peer review reports and author responses from that submission.

Round  1

Reviewer 1 Report

The manuscript is presenting an interesting subject, but unfortunately, the paper is hard to follow as it needs extensive English language improvements.

The authors are kindly requested to clarify if they are discussing "adsorbated films" or "absorbate film".

For the convenience of the readers, Section 2 needs rearrangement, as in the present form is hard to follow and to understand the need of the presented data.

Please clarify if the presented data in Table 1, are all from reference [4].

Please be more specific about the expression "a tiny enough lateral tensile stress" (line 207).

For the readers' convenience please consider the presentation of the AFM figure from [29].

Author Response

Comment 1: The manuscript is presenting an interesting subject, but unfortunately, the paper is hard to follow as it needs extensive English language improvements. 

  Response: Many thanks for this good comment. More efforts have been made to improve English language.

Comment 2: The authors are kindly requested to clarify if they are discussing "adsorbated films" or "absorbate film". 

  Response: We replaced "absorbate film" into “adsorbed films”, and similar terms have been changed.

Comment 3: For the convenience of the readers, Section 2 needs rearrangement, as in the present form is hard to follow and to understand the need of the presented data. 

  Response: We have rearranged Section 2 according to the Reviewer’s comments (Line 86-89, 100-105, 137-139,145-147, 176-178).

Comment 4: Please clarify if the presented data in Table 1, are all from reference [4]. 

  Response: We have re-written this part according to the Reviewer’s suggestion (Line 189-191). Some incorrect symbols of the experiment data have also been corrected (Table 1).

Comment 5: Please be more specific about the expression "a tiny enough lateral tensile stress" (line 207).

Response: We have replaced the vague statement of “tiny enough” as “a tiny lateral tensile stress far less than the residual stress induced by pre-stretching” (Line 257-258).

Comment 6: For the readers' convenience please consider the presentation of the AFM figure from [29].

  Response: Thanks for the suggestion. We have made some improvements (Figure 1a; Line 86-88).

Reviewer 2 Report

A horribly flawed paper Reject

From the very first sentence one is confronted by a lack of competence in the subject area - Eukaryote DNA folding involves Chromatin whereas Prokaryote involves simple winding. The concentration of Sodium used is false - 50mM in the intracellular compartments hence those results have no sense. Why spermidine and not a DNA binding anti-biotic? Why the Co ammonium complex and not cis-platinum. 

More seriously there are cations present in the cell or bacterium, Na; K, Mg and Ca and all at present at different concentrations depending on localising in the Montasser paper; 

Direct measurement of the mechanism by which magnesium specifically modifies the mechanical properties of DNA.

By: Montasser, I; Coleman, A W; Tauran, Y; et al.

Biomicrofluidics  Volume: 11   Issue: 5   Pages: 051102   Published: 2017-Sep

it was clearly demonstrated that it is necessary to work at all concentrations

The literature is to say the least fragile

Reject

Author Response

Response: Thanks for this comments. Considering the reviewer’s suggestion, we have made some improvements (Line 18-20, 41-42, 74-75, 216-218, 230-231, 250-251). We agree with reviewer’s points in some degree. However, we have some different views.

In this paper, we focus on revealing the underlying physical mechanism of the surface elastic properties (e.g. the tensile surface stress and negative elastic modulus for attraction-dominated DNA films) and the resultant detection signals of sensors. For this purpose, we compared the results of the repulsion-dominated DNA film with the attraction-dominated film. Furthermore, in multivalent solution (e.g. sp6+), the predicted tensile surface stress of the attraction-dominated DNA film reveals the possible physical mechanism of the condensation found in organism. It should be mentioned that all the parameters used in this paper is based on the following experiments in vitro. And the related predictions agree well with the Legay’s and Domínguez’s AFM experiments as well as Zhang’s theoretical prediction. In addition, to supplement the experiment measurements for studying DNA condensation, we quoted the work of micro-fluidic coupled SNTs based titration experiments done by Montasser et al.

H.H. Strey; V.A. Parsegian; R. Podgornik, Equation of state for DNA liquid crystals: fluctuation enhanced electrostatic double layer repulsion. Phys. Rev. Lett. 1997, 78, 895-898; DOI: 10.1103/PhysRevLett.78.895.

B.A. Todd; V.A. Parsegian; A. Shirahata; T.J. Thomas; D.C. Rau. Attractive forces between cation condensed DNA double helices. Biophys. J. 2008, 94, 4775-4782; DOI: 10.1529/biophysj.107.127332.

Round  2

Reviewer 1 Report

The authors were requested to provide a figure with the experimental data obtained using the AFM. In the figure 1A is presented just the scheme of the AFM measurement so that the request was not well addressed.

Section 2 was not rearranged as it was suggested. 

Reviewer 2 Report

The problem remains with this manuscript, the authors have not carried out the correct experiments. The choice of ions is incorrect - Mg, Ca, Zn would be appropriate. The work requires expanding not simply rephrasing

Reject